# Comparative Mitogenomics Reveals Cryptic Species in *Sillago ingenuua* McKay, 1985 (Perciformes: Sillaginidae)

**DOI:** 10.3390/genes14112043

**Published:** 2023-11-04

**Authors:** Tianxiang Gao, Yijia Shi, Jiaguang Xiao

**Affiliations:** 1School of Fishery, Zhejiang Ocean University, Zhoushan 316022, China; gaotianxiang0611@163.com; 2Fisheries College, Jimei University, Xiamen 361021, China; eja0313@163.com; 3Third Institute of Oceanography, Ministry of Natural Resources, Xiamen 361005, China

**Keywords:** *Sillago ingenuua*, cryptic species, morphological consistency, genome features, genetic divergence

## Abstract

It is unreliable to identify marine fishes only by external morphological features. Species misidentification brings great challenges to fishery research, resource monitoring and ecomanagement. *Sillago ingenuua* is an important part of commercial marine fishes, and in which, the morphological differences between different groups are not obvious. Here, we compared different geographical groups of *S. ingenuua* which were collected from Xiamen, Dongshan, Keelung, Songkhla and Java. The results showed that all samples of *S. ingenuua* were similar in external morphological characteristics and the shape of the swim bladder, but there were two distinctive lineages which were flagged as cryptic species based on DNA barcoding. The comparative mitogenomic results showed that *S. ingenuua* A and *S. ingenuua* B were identical in structural organization and gene arrangement. Their nucleotide composition and codon usage were also similar. A phylogenetic analysis was performed based on 13 concatenated PCGs from eight *Sillago* species. The results showed that the genetic distance between *S. ingenuua* A and *S. ingenuua* B was large (D = 0.069), and this genetic distance was large enough to reveal that *S. ingenuua* A and *S. ingenuua* B might be different species.

## 1. Introduction

The circular mitochondrial DNA (mtDNA) of metazoans has a series of advantages, such as maternal inheritance, absence of recombination, higher mutation rate than nuclear DNA, etc. There are many appropriate genetic markers in mtDNA for population, phylogenetic and biogeographic studies [1,2]. The most popular marker cytochrome c oxidase subunit I gene (COI) has been proposed for ”DNA barcoding” for species identification [3]. As the powerpacks of eukaryotic cells, mitochondria are highly efficient at generating ATP [4]. This is attributed to the minimalist structure of mtDNA; after the Genome Reductive Evolution (GRE) process, many genes were lost or transferred to the nucleus [5,6,7]. Finally, mtDNA became a very lean double-stranded circular molecule which usually contains 13 protein-coding genes (PCGs), 2 ribosomal RNA genes (rRNAs), 22 transfer RNA genes (tRNAs) and a noncoding control region (CR) in fish [8].

*S. ingenuua* McKay, 1985, was first identified and named by taxonomist Roland McKay in his book on the Sillaginidae family. The holotype was collected from Chantaburi Gulf of Thailand in 1975 [9]. *S. ingenuua* is an important inshore commercial fish species around its wide distribution range including the coasts and estuaries of northern Australia, Thailand, India and China. Before McKay, *S. ingenuua* had been misidentified as *Sillago argentifasciata* Martin and Montalban, 1935, because the fin ray counts and lateral line scale counts of *S. ingenuua* agree to some extent with those of *S. argentifasciata* [10,11]. However, the absence of a well-defined silvery mid-lateral band, the ctenoid upper check scales and the smaller eye of *S. ingenuua* suggest that they are distinct species [9]. The swim bladder of *S. ingenuua* is similar to that of *Sillago ciliata* Cuvier, 1829, and *Sillago analis* Whitley, 1943, but the lateral line scale counts and vertebrae counts are quite different [9].

In this study, we integrated both morphological and molecular methods to explore whether *S. ingenuua* with its patchy distribution contains different lineages. We analyzed the mitogenomes of different lineages of *S. ingenuua* to explore the divergence between them. A phylogenetic analysis was conducted based on PCGs to gain insight on their phylogenetic status in the genus *Sillago* Cuvier, 1817.

## 2. Materials and Methods

### 2.1. Sample Collection

The whole fish specimens of *S. ingenuua* were collected from Dongshan (Fujian, 28 individuals), Xiamen (Fujian, 12 individuals), Keelung (Taiwan, 10 individuals) and Java (Indonesia, 30 individuals) on April 2014, November 2015, July 2014 and March 2015, respectively (Figure 1). Eight tissue samples were collected from Songkhla (Thailand) on December 2012 (Figure 1). Moreover, two *S. ingenuua* COI sequences (EF609469 and FJ155368) were downloaded from GenBank for phylogenetic reconstruction in this study. Dorsal muscle tissue samples were taken and preserved in 100% ethanol for DNA extraction. All whole fish specimens and tissue samples were stored in a freezer at −20 °C.

### 2.2. Morphological Analysis

The whole fish specimens were checked including their external morphology, vertebra and the shape of the swim bladder. The external morphological characteristics usually consists of counts and measurements such as fin rays, scales, gill rakers, head length, eye diameter, etc. [12]. General abbreviations of the external morphological characteristics used in this paper were SL, standard length; TW, total weight; A, the number of anal fin rays; D, the number of dorsal fin rays; V, the number of pelvic fin rays; C, the number of caudal fin rays; and P, the number of pectoral fin rays. After measurements were taken, the gill rakers on the first gill arch and the number of vertebrae were counted, and the shape of swim bladder was checked based on the anatomical process. The definition of the modified vertebrae followed McKay [12] and the terminology of appendages of the swim bladder followed Shao et al. [13] and McKay [12]. All measurements were made with dial calipers to the nearest 0.1 mm and weight values were estimated to the nearest 0.1 g. All experiments were carried out in accordance with the Laboratory Animal Management and Ethics Committee of the Third Institute of Oceanography (Ministry of Natural Resources).

### 2.3. DNA Extraction and PCR Amplification

Total genomic DNA was isolated from the muscle tissue by proteinase K digestion followed by the standard phenol/chloroform method [14]. Employing specific universal primers L5956 (5′-CACAAAGACATTGGCACCCT-3′) and H6558 (5′-CCTCCTGCAGGGTCAAAGAA-3′) [15], a partial sequence of the mitochondrial COI gene was amplified in a reaction mixture containing 17.5 μL of ultrapure water, 2.5 μL of 10× PCR buffer, 2 μL of dNTPs, 0.15 μL of Taq polymerase, 1 μL each of the DNA template and two primers. A TAKARA thermal cycler was used for PCR amplification; the basic settings were initial denaturation 5 min at 95 °C, denaturation 45 s at 94 °C and 35 cycles, annealing 45 s at 50 °C, extension 45 s at 72 °C, and a final extension 10 min at 72 °C. The purified product was used as the DNA template for cycle sequencing reactions, using a BigDye terminator cycle sequencing kit. An ABI Prism 3730 automatic sequencer (Applied Biosystems, Foster City, CA, USA) was used for bi-direction sequencing with the same primers used for PCR amplification.

The complete sequences of the mitochondrial genome of two *S. ingenuua* lineages were amplified using the long-PCR technique and primer-walking method in this study [16]. All PCR primers were designed and implemented in Primer Premier 5.0 software (PRIMER Biosoft International) based on congeneric sequences download from GenBank. Long-PCR and normal PCR reactions were finished by a TAKARA thermal cycler. All fragments were sequenced on an ABI Prism 3730 from both strands after purification.

### 2.4. Genetic Analysis

All sequences were edited and aligned using DNASTAR 7.1 software (DNASTAR, Madison, WI, USA) with the default parameters, and refined manually. The final aligned COI sequences were edited to 583 bp for the following analysis and submitted to GenBank (KU051978-KU051987, KU051989-KU052003 and MF958497-MF958501). Two complete mitochondrial genome sequences of *S. ingenuua* were also submitted to GenBank (MF958502 and MF958503). Pairwise genetic distances were analyzed in MEGA X [17], and then we constructed a neighbor-joining (NJ) tree under the Kimura 2-parameter (K2P) model. For the complete sequences of the mitochondrial genomes of *S. ingenuua*, all PCGs and rRNAs were identified using MITOS [18]. The codon usage and base composition of 13 PCGs were analyzed in MEGA X. AT skew and GC skew were confirmed according to the equations [19]. The cloverleaf secondary structures of all 22 tRNAs were identified using tRNAscan-SE (http://lowelab.ucsc.edu/tRNAscan-SE/, accessed on 10 May 2020). CR was determined by comparing with the homologous sequences.

Phylogenetic trees were constructed based on 13 concatenated PCGs of eight *Sillago* species including *Sillago aeolus* Jordan & Evermann, 1902; *Sillago asiatica* McKay, 1982; *Sillago indica* McKay, Dutt & Sujatha, 1985; *Sillago japonica* Temminck & Schlegel, 1843; *Sillago sihama* (Fabricius, 1775); *Sillago sinica* Gao & Xue, 2011; and two *S. ingenuua*, *Larimichthys crocea* (Richardson, 1846) (NC_011710) [20] and *Terapon jarbua* (Fabricius, 1775) (NC_027281) [21] were used as outgroups. Nucleotide sequences were aligned and edited using Clustal X 2.0 under the default settings [22]; all gaps and stop codons were removed and all sequences were concatenated into a sequence matrix (every sequence was 11,415 sites in length). The maximum-likelihood (ML) tree was built using PAUP* 4.0 [23] and the Bayesian inference (BI) tree was built using Mrbayes 3.12 [24]. The substitution model was selected using jModelTest 2 [25] in the Akaike Information Criterion (AIC) algorithm [26]. The ML analysis was estimated after 1000 bootstrap replicates under the GTR + I + G model. The Bayesian analysis used a set of optimal models (GTR + I + G for the 1st, 2nd and 3rd positions, respectively). Four Markov chains were run for 1,000,000 generations by sampling the trees every 100 generations. The first 25% of trees was discarded and the Bayesian posterior probabilities (BPP) were estimated based on the remaining 75% trees to finally obtain the consensus tree.

## 3. Results and Discussions

### 3.1. Taxonomy and Cryptic Diversity

In this study, all specimens that were compared were recognized as the same species. The external morphological characteristics of these specimens agreed with the original description of *S. ingenuua* described by McKay: body and head is pale brown to light fawn; there is no silvery mid-lateral band; dorsal fins and anal fin are almost hyaline with sparse black spots; the base of the pectoral fin is silver; and the caudal fin is forked with a grayish brown margin posteriorly. And, more remarkably, the body scales of *S. ingenuua* are susceptible to damage. Table 1 shows the comparative results of the countable properties of the samples in this study and the type specimens [9]. These measurements were considered to be consistent with negligible diagnostic value. The first dorsal fin of *S. ingenuua* had XI~XII (mostly XI), the second dorsal fin had I and 16~17 (mostly 17) soft fin rays; the anal fin rays had II and 16~18 (mostly 17) soft fin rays; the scale counts on the lateral line were 64~70, the scale counts above the lateral line were 3 or 4; the gill rakers were 3~4 + 8~10; vertebra: abdominal 13, modified 7~11, caudal 9~13, and total 33. The shapes of the swim bladders of samples in this study were similar to the sketching given by McKay [12] (Figure 2), with a short median anterior extension and five short, pointed anterolateral projections; the anterior two on each side projected almost laterally, while the posterior ones pointed posterior and laterally; there was a single tapering posterior extension; and a duck-like process was present ventrally.

DNA barcoding is widely used in species identification; it is an effective and accurate way to avoid the requirement for intricate taxonomical expertise from researchers or students [3]. It is refreshing that DNA barcoding is often used to discover new species or cryptic species, which have identical or similar morphological or ecological characters with its sibling species, but are very different based on DNA barcoding [27,28]. In the present study, the NJ tree (Figure 3) revealed that all the previously recognized *S. ingenuua* were split into two significant lineages (namely, *S. ingenuua* A and *S. ingenuua* B) with a 5.0% genetic distance between them. The *S. ingenuua* A lineage included samples from Dongshan, Xiamen, Java and Songkhla; the *S. ingenuua* B lineage included samples from Keelung and the two downloaded COI sequences which were from Taipei and Western Australia, respectively. Based on either the 10× or 2% rule [29,30], these two lineages bordered on provisional species statuses; alternatively, a sibling species could be concealed in the synonymy of *S. ingenuua*.

### 3.2. General Features of the Mitogenomes

The mitogenomes of *S. ingenuua* A and *S. ingenuua* B were 16, 449 bp and 16, 445 bp, respectively (Figure 4, Table 2). *S. ingenuua* had the shortest mitogenome compared with other *Sillago* species (Table 3). The factor causing these differences in length was primarily the variations in CR. The structural organization and gene arrangement of these *Sillago* fishes were identical (Figure 4). Both *S. ingenuua* A and *S. ingenuua* B had strand-specific biases in their gene compositions, with most genes encoded on the H-strand except ND6 and 8 tRNAs (Ala, Asn, Cys, Gln, Glu, Pro, Ser (UCN) and Tyr), which are encoded on the L-strand. This trait is very common in the mitogenomes of other vertebrates [31,32,33,34,35].

The AT content of the mitogenomes varied among *Sillago* taxa (Table 3) from 51.3% (*S. sinica*) to 54.5% (*S. ingenuua* A). This bias in nucleotide composition is universal in the mitogenomes of *Sillago* species, except for the first codon positions of PCGs [36,37,38]. For the GC /AT skew analyses, both *S. ingenuua* A and *S. ingenuua* B exhibited some distinctiveness. All *Sillago* species displayed a negative AT skew from −0.009 (*S. aeolus*) to −0.033 (*S. indica*), except for *S. ingenuua* (0.041 and 0.048 in *S. ingenuua* A and *S. ingenuua* B, respectively), and a strong negative GC skew from −0.190 (*S. sinica*) to −0.284 (*S. ingenuua* A) (Appendix A).

### 3.3. Protein-Coding Genes and Codon Usage

All 13 PCGs were also present in the mitogenomes of *S. ingenuua* A and *S. ingenuua* B, including the seven subunits of the NADH ubiquinone oxidoreductase complex (ND1-6, ND4L), three subunits of the cytochrome c oxidase (COI-III), one subunit of the ubiquinol cytochrome *b* oxidoreductase complex (Cyt *b*), and two subunits of the ATP synthases (ATP6 and ATP8) (Figure 4, Table 2). After removing the stop codons, the total length of the 13 concatenated PCGs were equal in length (11,400 bp) except for *S. japonica* which was 11,409 bp in length (Table 3). The mitogenomes of *S. ingenuua* A and *S. ingenuua* B exhibited the canonical genetic code shared by most vertebrates [39]. All PCGs used ATG as the orthodox initiation codon except for COI which used GTG as the start codon (Table 2). Table 2 shows the detailed usage record of stop codons including complete and incomplete stop codons, which is a common tendency in fish mitogenomes [33,37].

In total, except for the stop codons, 3800 codons were found in the mitochondrial genomes of *S. ingenuua* A and *S. ingenuua* B. Among them, codons for leucine were the most frequently used codons (17.45% and 17.47% in *S. ingenuua* A and *S. ingenuua* B, respectively). It was speculated that leucine plays a crucial part in encoding many transmembrane proteins of the chondriosome [40]. The codons for Cysteine were the least frequently used codons with a percentage value of 0.61%. The results for the relative synonymous codon usage (RSCU) showed that A-terminal and C-terminal codons were in great abundance and G-terminal codons were extremely sparse in the H-strand among the *Sillago* species (Figure 5). The asymmetrical directional mutation pressure may be the underlying mechanism responsible for the strand bias; this pressure was associated with the replication processes [19].

### 3.4. Transfer and Ribosomal RNA Genes

There are 22 tRNAs in mitogenomes of *S. ingenuua* A and *S. ingenuua* B, respectively, with 14 tRNAs on the H-strand and 8 tRNAs on the L-strand (Table 2). Most of tRNAs could be folded into the typical cloverleaf secondary structure (Appendix A). tRNASer (AGY) lacked the recognizable DHU stem, which is common in almost all vertebrate mitogenomes [41,42]. There were stem mismatches in the tRNAs of *S. ingenuua*, and this might be related to the post-transcriptional editing process [43].

Two rRNAs were found in *S. ingenuua* A and *S. ingenuua* B, respectively. The 12S subunit of rRNA was 947 bp and 948 bp in length and the 16S subunit was 1694 bp and 1693 bp, respectively. The sizes of the rRNAs were similar to those of other *Sillago* species (Table 3). Additionally, there were 7 bp mutation sites in 12S and 25 bp mutation sites in 16S between *S. ingenuua* A and *S. ingenuua* B.

### 3.5. Control Region

The mitochondrial genome control region is non-coding, AT-rich and has a high evolutionary rate [44]. The length of the CR varies, and this is the primary reason leading to the length variations in mitogenomes, but its control elements related to regulatory functions are known to be highly conserved [45,46,47]. The CR of *S. ingneuua* was located between the tRNAPro and tRNAPhe genes, and was determined to be 792 bp in *S. ingenuua* A and 787 bp in *S. ingenuua* B. *S. ingenuua* had the shortest CR among all the sequenced *Sillago* species (Table 3). No tandem repeat was detected in the CR of *S. ingenuua*.

The structures of the CR for *S. ingenuua* A and *S. ingenuua* B were identical. The CR could be divided into three domains (the termination associated sequence domain (TAS), Domain I; the central conserved sequence block domain (central), Domain II; and the conserved sequence block domain (CSB), Domain III) (Figure 6). In Domain I, the termination-associated sequences (TAS, 5′-TACAT-3′ and 5′-ATGTA-3′) are considered to be a signal to terminate the synthesis of the CR [48]. In Domain II, four conserved sequence boxes (F, E, D and C) were detected after aligning, which is consistent with other fish mitogenomes. In Domain III, the conserved sequence block domains CSB1, CSB2 and CSB3 were detected which are thought to be involved in positioning RNA polymerase both for transcription and priming replication [49].

### 3.6. Phylogenomic Relationships of Eight Species in Genus Sillago

Maximum-likelihood and Bayesian inference analyses were conducted with the concatenated nucleotide data of eight *Sillago* sequences and the two outgroup taxa. The topological relationships of the two phylogenetic analyses remained consistent, and all analyses provided high bootstrap support values for all internodes (Figure 7). The resulting topology showed two *S. ingenuua* lineages which exhibited obvious genetic differences, clustered as sisters of all other species of the genus *Sillago*. *S. aeolus* was recovered as a sister to *S. sinica*, *S. indica* was recovered as a sister to *S. sihama* and then they were recovered as sisters to *S. asiatica* and *S. japonica* and in the main clade. In accordance with the phylogenetic analyses, the genetic distance between *S. ingenuua* A and *S. ingenuua* B also revealed an obvious genetic differentiation (0.069) based on the 13 PCGs (Table 4). The mitogenomic data supported the deep intraspecific differentiations in *S. ingenuua*, revealing that *S. ingenuua* A and *S. ingenuua* B might be different species.

## 4. Conclusions

In summary, we suggested the existence of two lineages in *S. ingenuua* which are morphologically almost indistinguishable: one, denoted as type A, is distributed in mainland China, Thailand and Indonesia; the other, denoted as type B, is distributed in Taiwan and northern Australia. The presence of the Wallacea oceanic trough between Sundaland and Sahulland during the Pliocene and Pleistocene may provide an explanation for the differentiation between the China/Thailand/Indonesia clade and Australian clade [50]. However, an unexpected Taiwan *S. ingenuua* population which clustered into the Australian clade (Sahulland) rather than the Sundaland clade was still puzzling. The genome structure, base composition and skew were similar between the two lineages of *S. ingenuua*. The tree topologies obtained in the present study were identical and statistically well supported by high bootstrap and posterior probability values. Therefore, the results suggested that the two lineages of *S. ingenuua* could be genetically distinct as different species.

## Figures and Tables

**Figure 1 genes-14-02043-f001:**
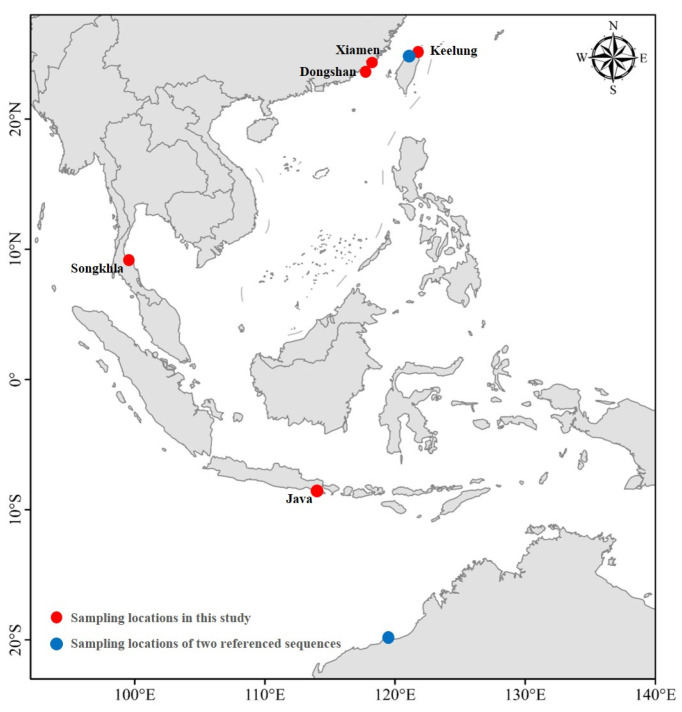
Sampling locations of *S. ingenuua*.

**Figure 2 genes-14-02043-f002:**
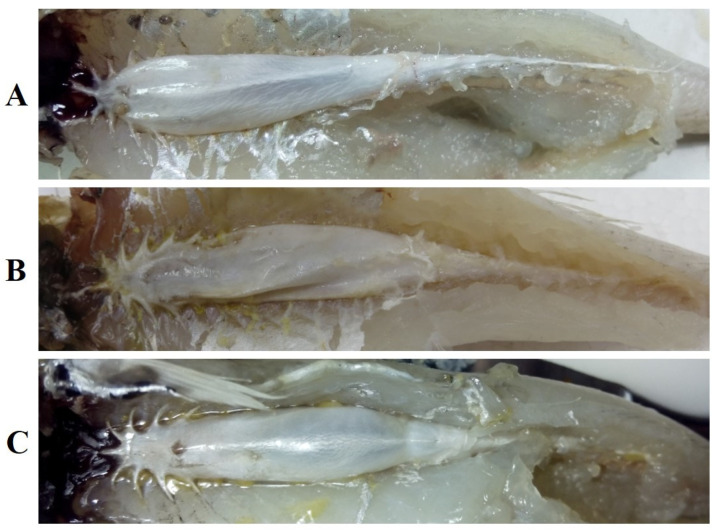
Swim bladders of *S. ingenuua* (collected from (**A**) Dongshan; (**B**) Java; (**C**) Keelung).

**Figure 3 genes-14-02043-f003:**
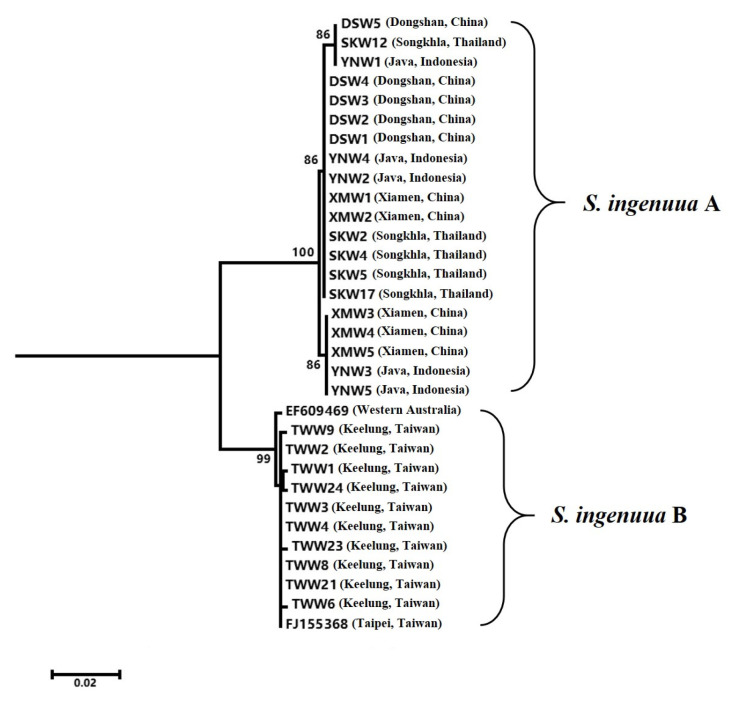
NJ tree for 32 COI sequences of *S. ingenuua*.

**Figure 4 genes-14-02043-f004:**
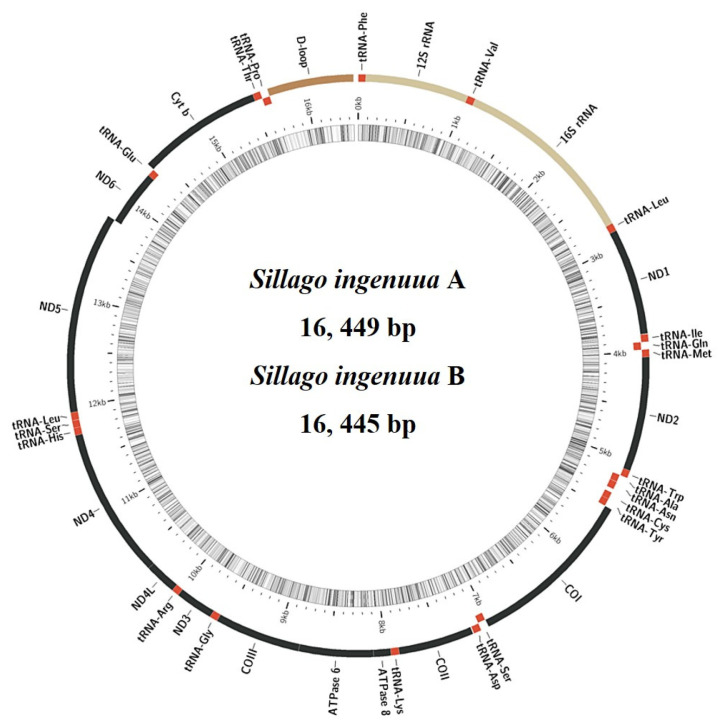
Gene organization for the mitogenomes of *S. ingenuua* A and *S. ingenuua* B. The structural organization and gene arrangement of *S*. *ingenuua* A and *S*. *ingenuua* B are identical; only the gene organization of *S*. *ingenuua* A is shown. Red regions represent tRNAs, yellow regions represent rRNAs, black regions represent PCGs and brown region represents CR.

**Figure 5 genes-14-02043-f005:**
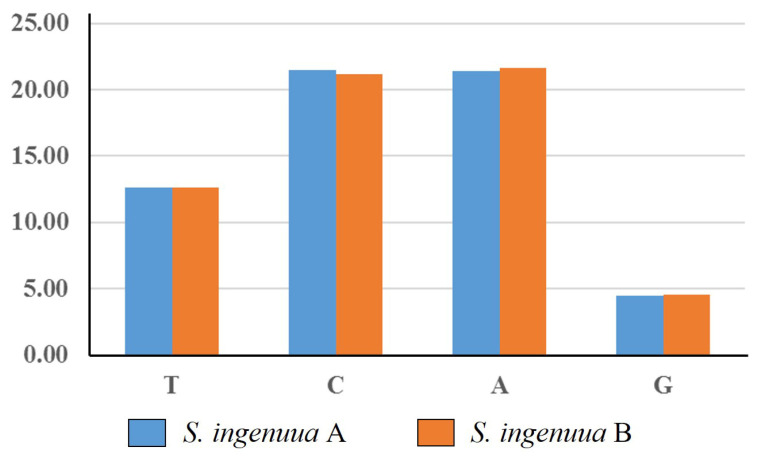
Frequencies of codons ending with the same nucleotide in H-strand. Y-axis represents the percentage of relative synonymous codon usage (RSCU) values of codons ending with the same nucleotide, and the X-axis represents all codon families.

**Figure 6 genes-14-02043-f006:**
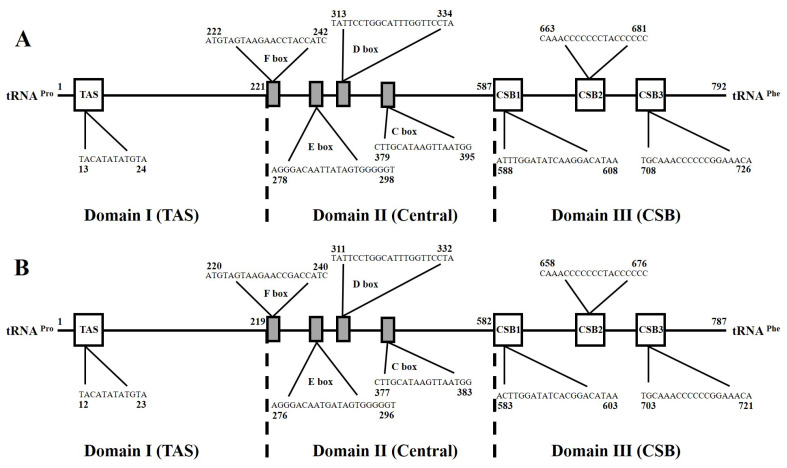
The main characteristics of mitochondrial control regions of *S. ingenuua* A (**A**) and *S. ingenuua* B (**B**). Termination-associated sequence (TAS), central conserved sequences (CSB-F, -E, -D, -C) and sequence blocks (CSB-1, -2, -3) were identified.

**Figure 7 genes-14-02043-f007:**
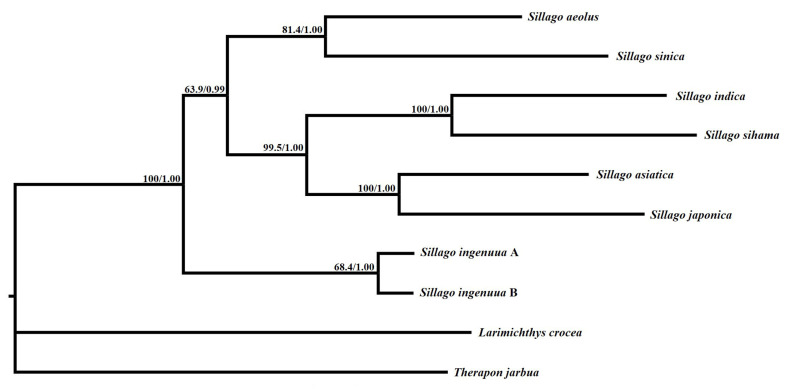
Inferred phylogenetic relationships among *Sillago* based on the concatenated nucleotide sequences of 13 mitochondrial protein-coding genes using maximum likelihood (ML) and Bayesian inference (BI). Numbers on branches are bootstrap percentages and Bayesian posterior probabilities. *L. crocea* (NC_011710) and *T. jarbua* (NC_027281) are outgroups.

**Table 1 genes-14-02043-t001:** Morphometric measurements of *S. ingenuua* samples.

	Dongshan	Xiamen	Keelung	Java	Chantaburi ^1^
TW (g)	15.7~35.4	20.6~35.9	24.31~37.27	37.7~60.2	-
SL (mm)	112.5~142.2	124.0~138.8	111.2~140.0	144.0~178.4	105.0~158.0
D	XI, I, 17	XI, I, 17	XI~XII, I, 16~17	XI, I, 16~17	XI, I, 17
A	II, 17	II, 17	II, 17	II, 16~18	II, 17
V	I, 5	I, 5	I, 5	I, 5	-
P	14~16	15~16	15~16	16	-
C	17~18	17~18	17~18	17~18	-
Scales on lateral line	64~68	65~69	66~68	65~69	66~70
Scales above lateral line	3~4	3~4	3~4	3~4	3~4
Scales below lateral line	8~9	8~9	8~9	7~9	8~9
Gill rakers	3~4/8~10	3~4/8~10	3~4/8~10	3~4/8~10	-
Vertebrae ^2^	13 + 7~11 + 9~13 = 33	13 + 8~10 + 10~12 = 33	13 + 9 + 11 = 33	13 + 8~10 + 10~12 = 33	13 + 9~11~9~11 = 33

^1^ Morphometric measurements for type specimens in McKay [9]; ^2^ the vertebrae are divided into 3 sections following by McKay [12]: abdominal vertebrae, hemal funnel and caudal vertebrae.

**Table 2 genes-14-02043-t002:** Characteristics of the mitochondrial genomes of *S. ingenuua* A and *S. ingenuua* B.

Gene/Region	Position		Size (bp) Nucleotide (A/B)	Amino Acid	Gap ^2^ (A/B)	Codon		Strand
	From (A/B)	To (A/B)				Start	Stop ^1^	
tRNA^Phe^	1	68	68		0			H
12S rRNA	69	1015/1016	947/948		0			H
tRNA^Val^	1016/1017	1087/1088	72		0			H
16S rRNA	1088/1089	2781	1694/1693		0			H
tRNA^Leu^(UUR)	2782	2855	74		0			H
ND1	2856	3830	975	324	4	ATG	TAA	H
tRNA^Ile^	3835	3904	70		−1			H
tRNA^Gln^	3904	3974	71		−1			L
tRNA^Met^	3974	4043	70		−1			H
ND2	4043	5088	1046	348	0	ATG	TA	H
tRNA^Trp^	5089	5159	71		0			H
tRNA^Ala^	5160	5229	70		1			L
tRNA^Asn^	5231	5303	73		0			L
O_L_	5304	5342	39		−3			H
tRNA^Cys^	5340	5405	66		0			L
tRNA^Tyr^	5406	5475	70		1			L
CO I	5477	7027	1551	516	0	GTG	TAA	H
tRNA^Ser^(UCN)	7028	7098	71		2			L
tRNA^Asp^	7101	7172	72		10			H
CO II	7182/7183	7872/7873	691	230	0	ATG	T	H
tRNA^Lys^	7873/7874	7946/7947	74		1			H
ATPase8	7948/7949	8115/8116	168	55	−10	ATG	TAA	H
ATPase6	8106/8107	8789/8790	684	227	−1	ATG	TAA	H
CO III	8789/8790	9573/9574	785	261	0	ATG	TA	H
tRNA^Gly^	9574/9575	9645/9646	72		0			H
ND3	9646/9647	9994/9995	349	116	0	ATG	T	H
tRNA^Arg^	9995/9996	10,064/10,065	70		−1			H
ND4L	10,064/10,065	10,360/10,361	297	98	−7	ATG	TAA	H
ND4	10,354/10,355	11,734/11,735	1381	460	0	ATG	T	H
tRNA^His^	11,735/11,736	11,803/11,804	69		0			H
tRNA^Ser^(AGY)	11,804/11,805	11,871/11,872	68		2			H
tRNA^Leu^(CUN)	11,874/11,875	11,946/11,947	73		0			H
ND5	11,947/11,948	13,785/13,786	1839	612	−4	ATG	TAA	H
ND6	13,782/13,783	14,303/14,304	522	173	0	ATG	TAG	L
tRNA^Glu^	14,304/14,305	14,371/14,372	68		4			L
Cyt *b*	14,376/14,377	15,516/15,517	1141	380	0	ATG	T	H
tRNA^Thr^	15,517/15,518	15,587/15,588	71		−1			H
tRNA^Pro^	15,587/15,588	15,657/15,658	71		0			L
Control region	15,658/15,659	16,449/16,445	792/787					H

^1^ TA and T represent incomplete stop codons; ^2^ positive numbers correspond to the nucleotides separating adjacent genes, negative numbers indicate overlapping nucleotides.

**Table 3 genes-14-02043-t003:** Genomic characteristics of eight *Sillago* mitochondrial genomes.

Species	GenBank Accession	Genome		13 Protein-Coding Genes					2 rRNA		22 tRNA		CR	
		Length (bp)	A + T (%)	Length (bp) ^1^	A + T (%)				Length (bp)	A + T (%)	Length (bp)	A + T (%)	Length (bp)	A + T (%)
					All Positions	1st Codon Position	2nd Codon Position	3rd Codon Position						
*S. ingenuua* A	MF958502	16,449	54.5	11,400	54.1	46.0	58.8	57.4	2641	53.7	1554	53.6	792	65.4
*S. ingenuua* B	MF958503	16,445	54.4	11,400	54.0	46.1	58.8	57.2	2641	53.3	1554	53.7	787	64.0
*S. aeolus*	NC025935	16,499	51.5	11,400	50.7	45.3	58.4	48.4	2638	51.5	1554	53.3	846	59.2
*S. asiatica*	NC025337	16,493	52.0	11,400	51.1	45.6	58.8	49.0	2644	51.9	1557	53.8	827	59.7
*S. indica*	NC025298	16,647	52.8	11,400	52.2	46.2	58.4	51.9	2654	52.2	1558	53.5	952	61.1
*S. japonica*	KR363149	17,119	53.9	11,409	53.2	46.3	58.8	54.4	2677	53.2	1566	54.0	1068	62.4
*S. sihama*	KR363150	17,003	52.3	11,400	51.5	45.3	58.5	50.8	2658	51.0	1557	53.4	1026 ^2^	60.5
*S. sinica*	NC030373	16,572	51.3	11,400	50.5	45.2	58.5	47.9	2648	52.1	1558	53.5	831	58.2

^1^ Excluding the stop codons; ^2^ redundant repeats were removed and only one unit kept for analysis.

**Table 4 genes-14-02043-t004:** Matrix of net average genetic distances based on 13 protein-coding gene sequences among the genus *Sillago*.

	*S. ingenuua* A	*S. ingenuua* B	*S. aeolus*	*S. asiatica*	*S. indica*	*S. japonica*	*S. sihama*
*S. ingenuua* B	0.069						
*S. aeolus*	0.234	0.231					
*S. asiatica*	0.257	0.258	0.269				
*S. indica*	0.265	0.262	0.281	0.265			
*S. japonica*	0.267	0.264	0.279	0.232	0.271		
*S. sihama*	0.266	0.272	0.277	0.269	0.234	0.280	
*S. sinica*	0.273	0.275	0.217	0.276	0.281	0.286	0.281

## Data Availability

The original contributions presented in the study are included in the article and Appendix A; further inquiries can be directed to the corresponding author.

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
