# Peer review of "Comparative Mitogenomics Reveals Cryptic Species in Sillago ingenuua McKay, 1985 (Perciformes: Sillaginidae)"

_genes, 2023, doi:10.3390/genes14112043_

Round 1
Reviewer 1 Report
Comments and Suggestions for Authors
Dear Authors,
I congratulate you for the work! I highly recommend you English proofreading of the manuscript. Concerning the results, I would rather say that more evidences are needed to support the conclusions. However, at the moment, based on the arguments provided, the results could be considered enough valid and ready for publication. You may try in the future other bioinformatic methods for phylogenetic predicion just in order to compare the results. I attach the document in which I made just a few comments.

I highly recommend you English proofreading of the manuscript!
Author Response
Dear reviewers:
Thank you very much for your comments to our manuscript entitled “Comparative mitogenomics reveals cryptic species in Sillago ingenuua McKay, 1985 (Perciformes: Sillaginidae)” (Paper ID: genes-2673146).
Those comments are all valuable and very helpful for revising and improving our paper. We have studied comments carefully and have made corrections which we hope meet with approval. Revised portion can be tracked in review Word markup version.
(1) We modified the text word for word, and nearly all sentences have been improved. All grammatical errors and other errors have been checked.
(2) In line 28, this sentence was rewritten, “comparatively high mutation rate” was replaced by “higher mutation rate than nuclear DNA”.
(3) In line 114, the software “DOGMA” was replaced by “MITOS”, and related reference was also replaced.
(4) In line 301-303, this part has been deleted.
(5) Other corrections see attached markup version.
On behalf of all authors.
Thank you very much.
Reviewer 2 Report
Comments and Suggestions for Authors
The paper is interesting and present good data. Some minor questions were marked in the text.

Overall the English is ok but can be improved.
Author Response
Dear reviewers:
Thank you very much for your comments to our manuscript entitled “Comparative mitogenomics reveals cryptic species in Sillago ingenuua McKay, 1985 (Perciformes: Sillaginidae)” (Paper ID: genes-2673146).
Those comments are all valuable and very helpful for revising and improving our paper. We have studied comments carefully and have made corrections which we hope meet with approval. Revised portion can be tracked in review Word markup version.
(1) We modified the text word for word, and nearly all sentences have been improved. All grammatical errors and other errors have been checked.
(2) In line 9, the first sentence has been rewritten as “It is unreliable to identify marine fishes only by external morphological features.”
(3) In line 20, this sentence has been rewritten as “this genetic distance was enough deep to….”.
(4) In line 30, we deleted the word “Typically”.
(5) In line 31, “cytochrome oxidase subunit I gene” was replaced by “cytochrome c oxidase subunit I gene”.
(6) In line 161, this figure was replaced by the photos of swim bladders.
(7) Other corrections see attached markup version.
On behalf of all authors.
Thank you very much.
Reviewer 3 Report
Comments and Suggestions for Authors
Dear authors,
The manuscript is a great contribution to the knowledge of the genus Sillago. However, there are many flaws present in the current text.
The authors and years of the species must always accompany the name of the species or genus the first time it is mentioned in the text.
The introduction talks about Sillago argentifasciata x S. ingenuua, however after that there is no further mention of this species.Why is she not in the analysis? Could it be one of the species analyzed? If the objective of the work was to explore the potential new species, the introduction should be better written so that it does not come as a surprise in the material and method.
The NJ analysis is somewhat unnecessary in the text, it can be placed in the supplementary material.
Regarding the EF609469 sequence from Australia, I would check the origin and quality of the sequence carefully as there seems to be something wrong with it (it doesn't make much sense about its position in the phylogeny.)
The words used to refer to clade relationships must be changed as in the revised PDF, as they are all wrong. Finally, the work has enormous potential but as it was presented I cannot say that it is suitable for publication without first having a good explanation of why Sillago argentifasciata is not in the analysis since S. ingenuua was previously identified as this species.

Author Response
Dear reviewers:
Thank you very much for your comments to our manuscript entitled “Comparative mitogenomics reveals cryptic species in Sillago ingenuua McKay, 1985 (Perciformes: Sillaginidae)” (Paper ID: genes-2673146).
Those comments are all valuable and very helpful for revising and improving our paper. We have studied comments carefully and have made corrections which we hope meet with approval. Revised portion can be tracked in review Word markup version.
(1) We modified the text word for word, and nearly all sentences have been improved. All grammatical errors and other errors have been checked.
(2) The authors and years of the species and genus have been added the first time it is mentioned in the text.
(3) About Sillago argentifasciata, Its veracity is questionable because there is no subsequent sample can be found and the type specimens were destroyed during the Second World War. This species is similar to Sillago ingenuua but we can easily distinguish them from the silvery mid-lateral band, the ctenoid upper check scales and the eye diameter. We added this part of explanations in Introduction.
(4) The NJ analysis is still retained in the main text, to show the distribution of different lineages of S. ingenuua.
(5) EF609469 sequence from Australia and FJ155368 sequence from Taipei are clustered together with our samples collected from Keelung, the accuracy of the data is OK. It's no doubt that there are two lineages of S. ingenuua.
(6) In line 2, the author and year of Sillago ingenuua has been added.
(7) In line 9, the first sentence has been rewritten as “It is unreliable to identify marine fishes only by external morphological features.”
(8) In line 18, “8” was replaced by “eight”.
(9) In line 19, this sentence was rewritten as “The results showed that the genetic distance between S. ingenuua A and S. ingenuua B was great (D = 0.069),”
(10) In line 31, “diagnosis” was replaced by “identification”.
(11) In line 39, the sentence was rewritten as “Sillago ingenuua McKay, 1985, was first identified and named by taxonomist Roland McKay in his book of the Sillaginidae family.”
(12) In line 40, the sentence was rewritten as “The holotype was collected from Chantaburi Gulf of Thailand in 1975”.
(13) In line 41, “S. ingenuua” was replaced by “Sillago ingenuua”.
(14) In line 43, “Prior to this official naming,” was replaced by “Before McKay,”.
(15) In line 44, the author and year of Sillago argentifasciata has been added. “S. ingenuua” was replaced by “Sillago ingenuua”.
(16) In line 55, the author and year of Sillago has been added.
(17) In line 73, “ventral” was replaced by “pelvic”.
(18) In line 116, the authors and years of these six Sillago species were added and this sentence was moved to next paragraph before outgroups.
(19) In line 126, the authors and years of Larimichthys crocea and Terapon jarbua have been added.
(20) In line 144, this sentence was rewritten as “The external morphological characteristics of these specimens agreed with the original description of S. ingenuua described by McKay:…”
(21) Figure 2 showed swim bladder, cannot see scale after dissection.
(22) In line 288, “8” was replaced by “eight”.
(23) In line 289, this first sentence was rewritten as “Maximum-likelihood and Bayesian inference analyses were…”
(24) In line 292-296, this part was rewritten as “The resultant topology showed two S. ingenuua lineages which exhibited obvious genetic differentiation clustered as sisters of all other species of the genus Sillago. Sillago aeolus was recovered as sister to S. sinica firstly, S. indica was recovered as sister to S. sihama and then they were recovered as sister to S. asiatica and S. japonica, they were recovered in the main clade.”
(25) In line 324, this sentence was rewritten.
(26) Other corrections see attached markup version.
On behalf of all authors.
Thank you very much.
Round 2
Reviewer 3 Report
Comments and Suggestions for Authors
Dear authors,
I believe that the manuscript has improved a lot and is ready for publication, you should just change in the title "Perciformes" to "Acanthuriformes", as the order has been changed for some time.